# Melatonin in Human Breast Milk and Its Potential Role in Circadian Entrainment: A Nod towards Chrononutrition?

**DOI:** 10.3390/nu16101422

**Published:** 2024-05-08

**Authors:** Silke Häusler, Emma Lanzinger, Elke Sams, Claudius Fazelnia, Kevin Allmer, Christoph Binder, Russel J. Reiter, Thomas K. Felder

**Affiliations:** 1Division of Neonatology, Department of Pediatrics, Paracelsus Medical University, 5020 Salzburg, Austria; emma.lanzinger@stud.pmu.ac.at (E.L.); e.sams@salk.at (E.S.); 2Department of Obstetrics and Gynecology, Paracelsus Medical University, 5020 Salzburg, Austria; c.fazelnia@salk.at; 3Department of Laboratory Medicine, Paracelsus Medical University, 5020 Salzburg, Austria; k.allmer@salk.at (K.A.); t.felder@salk.at (T.K.F.); 4Division of Neonatology, Pediatric Intensive Care Medicine and Neuropediatrics, Department of Pediatrics and Adolescent Medicine, Medical University of Vienna, 1090 Vienna, Austria; christoph.a.binder@meduniwien.ac.at; 5Department of Cell Systems & Anatomy, UT Health San Antonio, 7703 Floyd Curl Drive, San Antonio, TX 78229, USA; reiter@uthscsa.edu; 6Institute of Pharmacy, Paracelsus Medical University, 5020 Salzburg, Austria

**Keywords:** melatonin, human breast milk, neonate, preterm, nursing, light–dark cycle, circadian rhythm, proper development

## Abstract

Breastfeeding is the most appropriate source of a newborn’s nutrition; among the plethora of its benefits, its modulation of circadian rhythmicity with melatonin as a potential neuroendocrine transducer has gained increasing interest. Transplacental transfer assures melatonin provision for the fetus, who is devoid of melatonin secretion. Even after birth, the neonatal pineal gland is not able to produce melatonin rhythmically for several months (with an even more prolonged deficiency following preterm birth). In this context, human breast milk constitutes the main natural source of melatonin: diurnal dynamic changes, an acrophase early after midnight, and changes in melatonin concentrations according to gestational age and during the different stages of lactation have been reported. Understudied thus far are the factors impacting on (changes in) melatonin content in human breast milk and their clinical significance in chronobiological adherence in the neonate: maternal as well as environmental aspects have to be investigated in more detail to guide nursing mothers in optimal feeding schedules which probably means a synchronized instead of mistimed feeding practice. This review aims to be thought-provoking regarding the critical role of melatonin in chrononutrition during breastfeeding, highlighting its potential in circadian entrainment and therefore optimizing (neuro)developmental outcomes in the neonatal setting.

## 1. Introduction

Due to its numerous benefits for neonatal wellbeing, the WHO highly recommends breastfeeding for optimal neonatal and childhood development. The biological components responsible for the superiority of breastfeeding over formula feeding are the subject of intensive research. The development and refinement of circadian rhythmicity from a mother’s milk to the neonate has gained increasing interest in this context. Breast milk composition changes significantly over the day in terms of diurnal hormonal changes; one of the molecules that is intricately linked to and responsible for circadian rhythmicity is melatonin, with night-time milk showing higher melatonin levels than day-time milk [1]. During pregnancy, melatonin, a tryptophan-derived endogenous neurohormone, exerts a myriad of physiological and biological functions that account for a stable environment for both the mother and the fetus [2]. Besides influencing circadian rhythmicity, melatonin acts as a free radical scavenger with anti-apoptotic, anti-inflammatory, and mitochondrial-stabilizing functions [3,4,5]. Above that, melatonin exerts immunological protection by boosting phagocyte activity and lymphocyte survival in colostrum, underscoring its potential importance when dealing with infections (which might be of particular relevance in maternal obesity during the pre- and postnatal period) [6,7]. Throughout the prenatal period, the fetus is not capable of circadian melatonin production and is therefore dependent on maternal melatonin secretion via transplacental transfer [8,9]. The increase in melatonin levels in the mother at 32 weeks of gestation emphasizes even further the importance of melatonin in fetal and neonatal development. Special consideration should therefore be given to infants born prematurely since they are not exposed to the final stage of pregnancy when maternal melatonin supply to the fetus is the highest.

Rhythmic melatonin secretion by the pineal gland only starts two to three months after birth; thus, neonates may lack adequate amounts of melatonin during this postnatal period [10,11]. In preterm neonates, this period of circadian melatonin deprivation is even more prolonged, making them more vulnerable to chronobiotic dysregulation and inflammatory conditions [10,12] So, for several months after birth, neonates rely at least partly on exogenous melatonin supplied by mother’s breast milk [13,14,15,16].

There is great interindividual variability in breast milk melatonin content with regard to gestational age at birth (preterm versus term) as well as the stage of lactation [1,17,18]. The clinical significance of this variability remains to be determined. Additionally, it will be important to identify the impact of the circadian variations in melatonin supplied by breast milk, i.e., low levels during the day and higher values at night, in terms of their effect on newborn circadian regulation. Beyond that, there is an urgent need to recognize internal risk factors for disturbances in maternal circadian rhythmicity with ensuing perturbed melatonin content in breast milk; these factors include maternal age and conditions (e.g., mastitis) which may critically impact neonatal melatonin supply via breast milk [19,20]. Additionally, external risk factors such as shift work and bright light exposure during nurturing at night should be minimized to ensure circadian melatonin secretion in human breast milk, which may play a role in the development and maintenance of the circadian entrainment of the newborn after birth [18,21,22].

## 2. Circadian Rhythmicity during Fetal Development—Placental Supply

The importance of circadian rhythmicity during development is becoming more widely acknowledged. In humans, the circadian clock (central rhythm generation) relies on a master clock located in the suprachiasmatic nucleus (SCN) of the hypothalamus; it influences circadian gene expression in peripheral cells (peripheral oscillators) throughout the body. The behavior is driven by 24 h mechanisms that operate through transcription–translation feedback loops. These loops regulate the activity of specific clock genes at the SCN and also at the cellular level downstream [23,24,25]. Internal clocks are entrained by external zeitgebers, e.g., environmental cues (e.g., light/dark cycles), that can impact the pattern of clock gene expression and thus the phase of overt circadian rhythms [26,27]. Therefore, the overall metabolism and cell/tissue function are accordingly synchronized [28].

Throughout pregnancy, the fetus is exposed to the circadian cycles of the mother by which its physiological processes and metabolic balance are likewise coordinated. During this period, the fetal clock is entrained by maternal non-photic rhythmic signals, i.e., rest –activity pattern, body temperature, uterine motility, and hormonal levels, rather than by external stimuli [15,29]. As early as 30 weeks of gestation, the fetus shows rhythmic patterns in total activity, heart rate, and breathing [15,30]. By 32 weeks, distinct behavioral patterns based on heart rate and body and eye movements, suggestive of cycles of active and quiet sleep, have been described in utero as well as in infants born preterm [31,32].

Special consideration is placed on neurohumoral signals, most notably melatonin, as mediators of maternal–fetal circadian communication. This nocturnally secreted molecule is synthesized from serotonin in the pineal gland which is driven by neuronal efferents from the sympathetic nervous system that terminate in juxtaposition to pinealocytes. In this context, a central innervation of the pineal gland has also been discussed, the fibers originating predominantly from perikarya located in hypothalamic and limbic forebrain structures as well as from perikarya in the optic system, which has been less investigated thus far but could also be of significance in the evolution and entrainment of melatonin rhythmicity [33]. Via this neuronal network, light detected by the eyes inhibits pineal melatonin production, with darkness allowing for ample melatonin production and secretion; as a result, peak circulating melatonin levels occur during the night [34,35]. During normal human pregnancy, maternal melatonin levels increase progressively from 24 weeks of gestation onwards until term and return to non-pregnant levels during the immediate post-partum period [36,37]. As shown in schematic representations in the cited papers, maternal levels in the third trimester are around 3-fold higher than in the first trimester with, due to the different measurement methods used, variable levels in the pg/mL range. However, daytime levels of melatonin do not increase accordingly during the same period, which leads to an even more pronounced amplitude in melatonin rhythmicity as gestation advances [9,38]. Of importance is that the rise in maternal melatonin levels coincides with the evolution of circadian rhythms in the fetus. Melatonin receptors have been postulated to be present at 18 weeks of gestation in the fetal SCN and diffusely distributed in fetal tissue [39]. Additionally, there are receptor-independent actions of melatonin, e.g., direct free radical scavenging.

Since in the fetal pineal gland, melatonin may not be produced sufficiently [10,40,41]—the optimal amount of supply is still to be determined and probably with great interindividual variation—the fetus is dependent on maternal melatonin, which easily crosses the placenta to enter the fetal circulation [42]. In compromised pregnancies, such as those associated with gestational diabetes mellitus or preeclampsia, melatonin homeostasis between the mother and fetus may be disrupted, culminating in alterations in fetal programming mainly through epigenetic changes [43]. In addition to its role as a regulator of circadian rhythmicity, melatonin’s antioxidant and anti-inflammatory properties may be markedly reduced in high-risk pregnancies [42]. Thus, mediated by melatonin, the connection between the circadian clock system and the maternal immune system seems as crucial for the maintenance of pregnancy as for the prevention of pregnancy-related inflammatory disorders, such as gestational diabetes, preeclampsia, miscarriage, fetal growth restriction. and preterm birth [44,45,46].

It is worth mentioning that, in this context, exogenous melatonin has already been shown to be safe and effectively used therapeutically in two small clinical trials in complicated pregnancies and during lactation. In a randomized controlled trial investigating the effects of melatonin (10 mg daily from the 15th to the 33rd week of pregnancy) in pregnant women with hyperglycemia, a significantly better glycemic control was observed, without any safety concern in the neonate in postnatal follow-ups [47].

In another small study that aimed to harness the antioxidant property of melatonin as a treatment for preeclampsia, pregnant women with early-onset preeclampsia received 30 mg of melatonin daily from diagnosis until delivery—in this phase I trial, melatonin significantly extended the mean diagnosis-to-delivery interval by 6 days and decreased the need for antihypertensive therapy compared to a historical control group, while no adverse drug reactions in the mothers and neonates were reported [48].

Pending further clinical trial assessment, melatonin remains a highly promising therapeutic option with its numerous attributes that would make it particularly attractive in pregnancy-related inflammatory conditions, as listed above [49].

## 3. Circadian Rhythmicity after Birth—Breast Milk as Continuum for Melatonin Provision

After birth, the newborn is exposed to a myriad of environmental changes: the maintenance of circadian entrainment for a proper establishment of sleep/wake rhythmicity on the basis of a functioning neurohormonal axis seems mandatory but fragile and is thus prone to disruptions within the first 3–6 months of life [30,50]. After birth, maternal pineal melatonin availability, as a key hormone in proper entrainment, is abruptly interrupted. One important option regarding how the loss of placentally transferred melatonin may be at least partly compensated is via breast milk transfer [5,18]. Over the course of lactation, human milk changes its composition to ideally fit the newborn’s nutritional demands. Its composition also varies throughout the day. The circadian fluctuations in breast milk melatonin concentration may be an important external cue to transfer chronobiological entrainment from mother to child [1], where melatonin exhibits high levels during the night, with the acrophase consistently after midnight, and significantly lower levels during the day. In one highly cited study, reported melatonin concentrations in milk were beyond the limit of detection (<43 pmol/L) during the day but 99 ± 26 pmol/L during the night. In this study, individual levels of 24 h breast milk melatonin samples collected within 3 months after delivery also showed a pronounced daily rhythm [51,52,53].

To our knowledge, only the aforementioned study evaluated melatonin levels in milk and serum simultaneously, analyzing serum samples only one time in a small cohort 3–4 days after birth—night concentrations of 280 ± 34 pmol/L in serum and 99 ± 26 pmol/L in milk were reported. On average, the melatonin concentration in breast milk was shown to be about 35% of the serum concentration [52]. No data have been published thus far on maternal serum melatonin levels during the different stages of lactation to correlate with melatonin levels in breast milk in the course of lactation.

Melatonin rhythmicity with higher melatonin content at night is observed at each of the three stages of lactation—colostrum, transitional milk and mature milk—and is independent of the mode of delivery or whether the fetus was born term or preterm [53,54,55].

The only (crossover) study examining dynamic changes in breast milk melatonin throughout the course of lactation in preterm versus term-born infants revealed that preterm breast milk had a higher concentration of melatonin than term breast milk in the colostrum (28.67 pg/mL vs. 25.31 pg/mL, *p* < 0.022), transitional breast milk (24.70 pg/mL vs. 22.55 pg/mL), and mature breast milk (22.37 pg/mL vs. 20.12 pg/mL), again also confirming higher night time compared to daytime values [55].

To date, no studies have been performed longitudinally investigating melatonin concentrations in human milk in women with pregnancy-related complications; thus, impaired melatonin levels based on lower maternal serum melatonin concentrations might be purely speculative in this context.

The higher concentration of melatonin and melatonin precursors may play an important role in neonatal sleep promotion [30]. As opposed to formula-fed infants, breastfed infants display more regular nocturnal increases in 6-sulfatoxymelatonin, melatonin’s catabolic metabolite excreted in urine, an improvement in nocturnal sleep with better sleep efficiency, a longer and less fragmented sleep, and a lower incidence of infant colic [56,57,58]. A study that examined the effects of exclusively breastfeeding vs. breastfeeding mixed with formula feeding showed that solely breastfed infants achieved a circadian rest–activity rhythm significantly earlier (6th week in contrast to 12th week of age) than the mixed formula-and-breastmilk-fed babies [59]. These findings may at least partially be dependent on the melatonin content in human milk since, in the absence of breast milk availability, supplementation with exogenous melatonin or melatonin precursors (tryptophan-enriched formula) displayed improvements in sleep/wake rhythms. This action requires the formula to be enriched with melatonin under dark conditions, thus pointing towards its importance in chrononutrition [60,61]. Thus, night-time breastfeeding results in increased melatonin supply to the newborn which, in addition to the entrainment of the infant’s circadian rhythms, also provides for powerful antioxidant, anti-inflammatory, and immune regulatory effects.

Another important aspect of the mother’s milk with its numerous bioactive substances is its interaction with the gut commensal bacteria and the local immune system. Human milk microbiota may play a critical role in the establishment and maturation of the intestinal microbiome, ultimately influencing intestinal inflammation and gut health during early life [62,63,64]. Increasing awareness has been directed towards gut microbiota impacting on brain homeostasis via the gut–brain axis [65,66]. In a mouse model, sleep deprivation-induced alterations in gut microbiota led to a colonic microbiota disorder as well as microglia overactivation and neuronal apoptosis in the central nervous system (CNS). By changing the composition of the gut microbiota, and thus dampening the inflammatory CNS response, melatonin ameliorated the sleep deprivation-induced cognitive impairment [67]. Breastfeeding in this context might be a particularly interesting means of orally administered melatonin due to varying concentrations following a circadian pattern, thus constituting an important chrononutrient: breast milk melatonin may directly act on gut microbiota composition, on the one hand, and provide a possible cue for the transmission of circadian rhythmicity while potentially also promoting longer night and total sleep trajectories on the other hand [57,68,69,70].

Apart from the interaction through gut–brain crosstalk, in the CNS, melatonin may have direct neurotrophic potential. In neonates, an increased level of endogenous neuronal activation primarily happens during the REM sleep state [71,72]. Numerous studies have demonstrated greater white and gray matter volumes in infants who were breastfed, pointing towards the importance of nutritional differences [73]. Whether the enhanced melatonin supply during night-time breastfeeding enhances circadian-modulated REM sleep states and enhances neurogenesis is an interesting possibility but requires further investigation.

Finally, melatonin plays a complex and multifaceted role as a modulator of the immune system: the interplay between melatonin and immunological compounds in colostrum has been evaluated in recent studies, which seems of significance not only but especially in pregnancies complicated by maternal obesity, since a high body mass index during pregnancy is reported to induce inflammation and changes in immune response in the neonate [74]. It was shown that in obese women, there was a reduction as well as functional impairment in their colostral immune cells; in in vitro experiments, melatonin could restore phagocytotic function as well as lymphocyte proliferation and reduce apoptosis [6,7]. The colostrum of mothers with a pre-gestational high body mass index exhibited higher melatonin levels compared to the nonobese control group. It was speculated that colostral melatonin may thus confer immunological stabilization which may also be linked to protection against obesity in the child [6].

Additionally, local changes in melatonin concentrations in colostrum based on the different phases of inflammation may be another important means to the initiation as well as the resolution of an inflammatory response following an antigenetic challenge. This was shown in one study dealing with maternal mastitis: reduced night-time melatonin levels in the acute phase of inflammation were followed by enhanced local colostral phagocytic melatonin production, probably contributing to the shut-down phase of the inflammatory response [20].

### Importance of Mother’s Milk in Preterm Infant

As already mentioned, circadian melatonin secretion increases progressively during pregnancy [8]. Therefore, preterm birth marks a disruption of this developmentally important fetal melatonin supply as well as chronobiological entrainment. Additionally, infants born prematurely may lack adequate endogenous melatonin production during several additional months since the interval is further prolonged compared to term-born neonates; this delayed melatonin rhythm occurs because the neurological circuitry that controls pineal melatonin production in the newborn is immature [75]. Typically encountered postnatal complications in preterms, such as fetal distress or fetal growth restriction, induce an even more pronounced delay in the initiation of circadian pineal melatonin secretion [76].

Additionally, preterm infants are born with an immature, still highly developing brain and are, when admitted to the neonatal intensive care unit, repeatedly exposed to stressful procedures. Postnatal stress exposure may contribute to the altered programming of the brain, including neurohumoral systems such as the hypothalamic–pituitary–adrenal axis, with the development of the autonomic nervous system and the circadian system. These disturbances occur during a phase where synaptogenesis, neuronal migration, and myelination as critical factors for the development of neural circuits by themselves may lead to a prolonged delay in maturation, which may also hold true for the neural pathways linked to circadian entrainment [77,78,79]. Due to its melatonin content, breast milk may provide preterms a zeitgeber after an abrupt loss of diurnal oscillating melatonin supply. As a quasi-counterbalance, the mother’s milk for preterm infants includes even higher melatonin concentrations, with a greater amplitude between day and night-time milk and colostrum with the highest melatonin content [55]. Nonetheless, throughout their intensive care period, preterms are still prone to desynchronized feeding schedules and no cycled lighting [1,12,80] and may therefore lack chronobiological adherence and rhythmicity. Accordingly, chrononutrition, i.e., appropriately timed feeding schedules might be particularly reasonable, which holds true for every newborn but should be even more emphasized in preterms based on the previously described high risk factors. Thus, mothers pumping and expressing milk should probably be encouraged to provide their (preterm) infants ideally with breast milk in a close timely manner to the time it was extracted—i.e., daytime milk should be fed during the day- and night-time milk during the night, respectively, to keep their baby in circadian synchrony.

Figure 1 summarizes the provision of melatonin during pregnancy as well as the first months after birth.

## 4. Factors Affecting Melatonin Content in Human Milk Composition

### 4.1. Maternal Factors

There are substantial interindividual variations in milk composition among the mothers: age [19,81], obesity [6,82,83], pregnancy-related morbidities [84], delivery mode [54,85], and environmental light pollution [86], as well as modifiable factors, such as smoking [87], eating habits [88], and, to some degree, maternal mental health [89,90] may contribute to this variability.

Prenatally, gestational exposure to environmental stress induces fetal growth restriction (FGR) which may partly be due to mitophagy-based mitochondria destruction and the resulting cellular bioenergy deficit. It was shown that melatonin confers protection against environmental stress-impaired progesterone synthesis and fetal growth via suppressing ROS-mediated mitophagy in placental trophoblasts [91].

In pregnancies complicated by placental pathology and insufficiency, especially preeclampsia and gestational diabetes, there is an imbalance between the excessive production of reactive oxygen species (ROS) and defensive antioxidative mechanisms, such as maternal melatonin levels [92]. As recently revealed in a systematic review, the circadian pattern of melatonin secretion seems to be disrupted in these pathologies as shown by the reduced production of melatonin and lower expression of melatonin receptors [84,93]. This may in turn lead to significant changes in maternal dynamic circadian signals and thus to developmental programming effects and fetal circadian disruption [9]. Whether the maternal melatonin deficiency translates into a reduced melatonin level and/or a lack of circadian rhythmicity in the breast milk has yet to be tested.

Maternal obesity in pregnancy, as already mentioned earlier, is associated with immunological, nutritional, and hormonal changes in colostrum and human milk composition, culminating in an increased risk of childhood obesity [74,94,95,96]. Melatonin has been shown to exert a myriad of “anti-obesogenic” effects, thereby ensuring energy homeostasis and preventing the evolution of obesity and metabolic syndrome in the offspring [97,98]. The finding that the colostrum of adipose mothers exhibited higher melatonin levels requires further delineation as to whether a certain threshold may confer protection against childhood obesity in follow-up [6].

Maternal variables such as maternal age and gestational age may also modify the content of melatonin in human breast milk. A study examining these two variables concluded that the antioxidant content was inversely connected with the maturational stage, suggesting that lower gestational age was associated with higher antioxidant levels. Additionally, advancing maternal age was linked to decreased melatonin content in breast milk; in this study, only transitional and mature milk but not colostrum was investigated and a sample collection was performed prior to midnight, i.e., before the reported natural peak melatonin concentration [19]. Maternal infection might also impact melatonin concentrations in the breast milk. A study compared melatonin levels in day and night colostrum samples between healthy puerperal mothers and mothers with mastitis and confirmed that the healthy mothers presented a significant day/night rhythm of melatonin, whereas the latter, affected by inflammation, did not [20]. It was assumed that this was due to a pro-inflammatory (TNF-α) inhibition of endocrine melatonin production.

### 4.2. Environmental Factors

#### Light

Light/dark cycles represent the most important external cue for setting the central internal clock, the suprachiasmatic nucleus. This might be of particular relevance in the neonatal intensive care unit where the melatonin rhythm in human breast milk may be impacted by this unique environment. Over the last three decades, various studies have investigated the effects of either cycled light versus continuous bright light or near darkness conditions on short-term infant outcomes [reviewed and summarized in [78]. In general, cycled lighting seemed to be safer and possibly superior in terms of clinical outcome and the length of hospital stay, although the reported findings were not always consistent as summarized in the most recent Cochrane Review on this topic [99]. Two studies additionally investigated salivary melatonin levels and suggested regulating circadian melatonin rhythmicity with cycled lighting [80,100]. However, both investigations lacked specific information regarding a detailed feeding schedule, mainly whether mother’s milk administration was carried out in an express-timed or mistimed fashion.

Maternal shift work which markedly alters the normal changing light/dark cycle causes profound internal clock desynchronization and conflicting zeitgeber conditions. Thus, due to external zeitgeber desynchrony, a potential disruption of the circadian rhythm on breast milk melatonin levels occurs. In a related study, breast milk samples were collected from shift-working mothers during either the day or at night and grouped into different day- and night-time intervals. The authors reported a significant reduction in breast milk melatonin during the midnight to 6:30 AM time interval on subsequent night shifts [22]. It is possible that the resulting circadian misalignment in breast milk melatonin would also likely impact neonatal circadian development. Additional research on the longitudinal effects of maternal chronodisruption on melatonin content in breast milk is warranted to better understand how a desynchronized melatoninergic message may interact with the circadian system development in the neonate.

The effects of nocturnal artificial illumination on the melatonin content in cow’s milk have been investigated. In this study, the melatonin concentrations in the milk were compared in daylight samples vs. samples under different night illumination conditions—natural darkness vs. short wavelength artificial light at night (“night-illuminated environment”). As expected, the melatonin concentrations of night-milk samples from the dark-night group were significantly higher than those from cows exposed to light at night. Interestingly, night-illuminated conditions also had an impact on melatonin concentrations in the daytime samples where values were significantly higher when the animals were in darkness at night compared to when they were maintained under night-illuminated conditions. The findings emphasize the importance of avoiding artificial light during night-time to obviate perturbations in the circadian melatonin rhythm. Another interesting finding in this study was that cows in both groups presented a significant daily rhythmicity in their heart rate, suggesting that in the night-illuminated cows, synchronized feeding and milking time may function as “zeitgebers” [101]. The results of this investigation also underscore the importance of “chrono-functional milk”, naturally rich with melatonin for newborns, and especially for preterm infants, who receive melatonin in milk supplied by indirect breast feeding. Also, an important unresolved question is whether pumping milk impacts its melatonin composition. Previous work has shown that women’s pumping and milk storage practices vary greatly and that indirect breastfeeding changes milk microbiota [102,103]. It remains unknown whether those changes in milk components may potentially be correlated with the melatonin content on the one hand or whether the pumping and storage procedure per se, on the other hand, under variable environmental cues (light and temperature), may directly interfere with breast milk melatonin content. Based on what is known to date, nursing mothers should probably be encouraged to maintain a regular day/night rhythm, sleep in a dark room, and breast feed or pump under dim-light or red-light conditions to ensure maximal levels of melatonin in breast milk at night. Avoiding blue wavelengths of light that are present in polychromatic artificial light sources (and also in cell phones or laptops) during the night seems of special importance since these light wavelengths are maximally inhibitory to pineal melatonin synthesis.

To better predict and quantify the circadian system response to different light exposure contexts, it is important to further understand the role of spectral sensitivity in regulating the human circadian system; only then may dosing effects as well as the temporal influence of exposure, photometrical standards, and optimal practices be established [104]. Additional research is necessary to definitively determine the influence of light exposure on melatonin secretion/suppression and appropriately counsel breastfeeding mothers about lighting schedules.

### 4.3. Pasteurization and Storage

The impact of pasteurization on breast milk melatonin content requires further investigations as well. Donated human milk with its unique composition is often used to replace the mother’s milk when it is not or sparingly available in sufficient amounts. For safety issues, most human milk banks still require that donated milk be pasteurized before administration. The pasteurization of a mother’s milk is also carried out to prevent biological hazards. The most commonly used heat treatment is Holder pasteurization (i.e., milk heated to 62.5 °C for 30 min); in a recent study on breast milk samples collected at night, it was demonstrated that there was a significant reduction in melatonin levels following Holder pasteurization (mean ± standard deviation = 51.92 pg/mL ± 19.54 versus 39.66 pg/mL ± 13.05), irrespective of whether a rapid or slowly cooling process after pasteurization was performed [105,106]. To our knowledge, no studies on the melatonin content following other possible pasteurization techniques have been performed and deserve further investigation [107].

Concerning the storage/processing conditions, melatonin levels in human milk, at least when immediately frozen following pumping, proved stable after freeze-thawing for up to 24 h: milk samples were collected during the first 6 months of breastfeeding, immediately frozen, and then defrosted after 4 months. Melatonin levels in milk were not significantly different in a timeline up to 24 h after defrosting. Also, in this study, focusing on the stability of melatonin in human milk, circadian melatonin rhythmicity in breast milk was evident [108].

### 4.4. Chrononutrition

As already mentioned earlier and independent of the light/dark cycle, the timing of food intake can affect the circadian system. Adjusting and coordinating nutrition quality and intake with the individual’s biological clock to consume the optimal type and quantity of nutrient components at the correspondingly optimal time of day might be especially important in neonates where the circadian rhythm is still maturing and who are receiving the mother’s timed cues through breastmilk. The circadian hormonal variations in glucocorticoids (peak levels during the morning) and melatonin (peak levels during the night) in the breast milk and the balanced interplay between these may be a trigger for coordinated sleep and alertness phases in the neonate. Thus, a disruption in the circadian secretion of either of these, respectively, may possibly cause a disruption in the diurnal rhythmicity [1,30,52,109]. Again, this might be of even more critical relevance in infants born (extremely) preterm, in whom the peri- and postnatal stress following the state of prematurity may render them even more dependent on dynamically changing breast milk components. The almost exclusive practice of using expressed milk in intensive care units means that the infants are usually fed “circadian-mismatched” milk, i.e., milk fed to the newborn at a different time of day from when it is expressed [110,111]. The mistimed feeding practice may lead to chronodisruption, with, as already mentioned, an impact on the long-term health of the newborn. It is thus essential to further define the effect of circadian variation on breast milk components and their changes during the different stages of lactation.

Irrespective of gestational age, current human milk banking practices do not consider circadian variation in milk composition and batch milk expressed at different times of the day. This practice may expose newborns to tonic levels of time-keeping hormones like melatonin or cortisol over 24 h periods, not considering the potential importance of their circadian peaks and rhythmicity. The clinical significance of continuous exposure to tonic hormonal milk signals on circadian biology still requires a thorough investigation [110].

Figure 2 summarizes and graphically depicts factors that might influence melatonin levels in human breast milk.

## 5. Summary and Open Questions for Future Research/Directions

After birth, nursing is the most appropriate form of nourishing the neonate. Subject to dynamic changes in composition, beginning with the first milk, colostrum, through transitional and mature milk, human milk adapts to the demands of gestational age and postnatal age, respectively [17]. Coupled to the maternal clock, the existence of fetal circadian rhythms has been shown in utero to begin at about 30 weeks gestation. Breast milk is regarded as a zeitgeber and may provide an early determinant for the development of circadian rhythmicity in the immediate postnatal period. Synchronization to the external environment after birth with the maintenance of circadian rhythmicity may possibly be triggered by exposure to varying melatonin concentrations via breast milk. In this context, the timing of feeding in synchronization with the light/dark cycle, i.e., chrononutrition, is likely a critical external cue [15]. Disruptions to these signals occur during the stay in an intensive care unit and may alter the development of circadian rhythmicity and sleep/wake cycling, with a potential to impact health and well-being throughout life [32,112,113].

For the optimization of chronoadherence in the (preterm) newborn, a number of uncertainties still require investigation. Firstly, a rational approach to the relevance of environmental lighting cycles as external zeitgebers is warranted in the care of hospitalized infants receiving breast milk; this should be a consideration for directly breastfed newborns and even more so for neonates receiving expressed milk. Even though cycled lighting seemed to be the most favorable approach to optimize melatonin levels in milk [78], there were a number of methodological challenges in the studies reported thus far. Light intensities, spectra, sources, duration as well as scheduled or unexpected light exposures (during routine care and in the case of unexpected procedures) have to be considered to fully understand the importance of the light/dark environment. Also, the optimum daylight intensity in the intensive care unit remains to be determined [114]. Importantly, a consideration of the importance of changes in melatonin levels based on perturbations in light exposure is rare. Thus, melatonin levels correlated with more detailed information on sleep/wake cycling patterns as a potential surrogate marker for the “ideal” circadian entrainment and subsequent (neurodevelopmental) outcome may be of relevance in further studies.

Most neonates in NICUs are still routinely fed mistimed expressed milk and thus lack the circadian rhythmicity of enteral melatonin supply. The impact of this desynchronized feeding practice requires further examination to obtain a better understanding of potential short- and long-term consequences of a possibly dysregulated circadian biology, mediated by a desynchronized melatonin supply to the newborn. If possible, randomized controlled trials should evaluate whether providing neonates with circadian- as well as stage-of-lactation-matched expressed milk improves clinical outcomes. A simultaneous longitudinal measurement of day and night melatonin levels and evaluation of sleep/wake cycling may provide important information on (the evolution of) circadian rhythmicity. Additionally, several variables as possible confounders should be taken into account, e.g., recording the prevailing light/dark cycle along with a maternal sleeping routine, rooming-in, noise, and light conditions during night-time milk expression should be documented. Furthermore, protocols should include detailed information of milk collection including labeling with the exact time it was expressed as well as uniform storage and pasteurization protocols. Moreover, information on maternal age, underlying diseases, especially pregnancy-related diseases, and the mode of delivery, as well as neonatal gestational age and weight at birth, may be important determinants impacting the melatonin content in a mother’s breast milk.

Only recently was it reported that among extremely preterm infants, donor human milk feeding did not result in different 2-year neurodevelopmental outcomes compared with preterm formula feeding [115]. However, no information was provided as to whether matched milk according to the stage of lactation was used or whether mistimed feeding was undertaken, which seems probable since this was not otherwise mentioned. Considering the large number of potential confounders, the outcomes of the studies could differ from those reported.

Finally, the adequate/ideal amount of breast milk melatonin supply in supporting the proper development of an infant’s circadian rhythmicity, which may vary in a circadian manner, would be a highly desirable goal for future studies. This may probably vary based on genetic background, gestational age, and possible underlying pathologies in the mother and the newborn. These findings could provide the basis for considering external melatonin supplementation for the mother during nursing or pumping, but they could also be important for infants receiving formula nutrition when breastfeeding is insufficient or not possible. To conclude, the beneficial effects of chrononutrition via breast milk supply on neonatal chronobiological entrainment may at least partly be mediated by its varying melatonin content. The importance of “chronolactomics” [15] in this context should be better defined and introduced into clinical practice to recognize breast milk as the important circadian cue in the postnatal period for the optimization of circadian rhythmicity in the NICU environment. Therefore, it seems reasonable to encourage breastfeeding on demand and at any time. If not possible, precision medicine on NICUs should presumably provide a timely matched human milk supply with an optimally adjusted (melatonin) composition, thus ideally matched to the babies’ needs and metabolic as well as developmental demands in order to improve wellbeing and ensure proper development.

## Figures and Tables

**Figure 1 nutrients-16-01422-f001:**
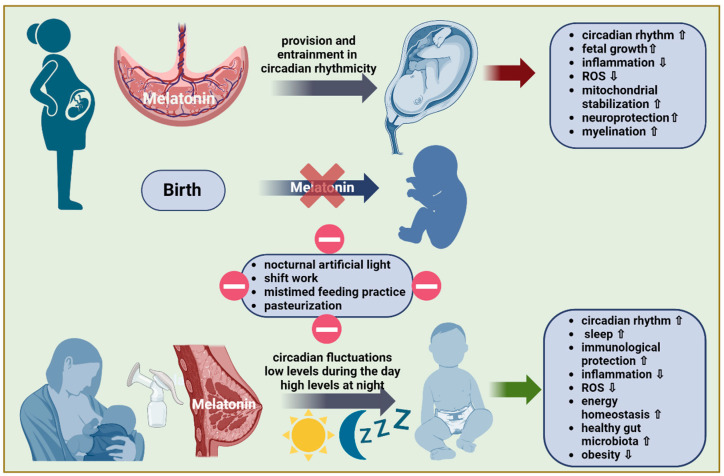
A schematic illustration of how melatonin supply from the mother to the fetus is provided throughout pregnancy and after birth. During pregnancy, maternal melatonin, expressed in a circadian fashion, easily crosses the placenta and significantly impacts the evolution of circadian rhythmicity in the fetus. Birth marks an abrupt interruption of the fetal melatonin supply. Breastfeeding is an important source of melatonin in the immediate postnatal period and during the first few months of life. Circadian fluctuations in melatonin concentration in breast milk with low levels during the day and high levels at night help maintain the newborn’s entrainment in circadian rhythmicity. Besides its function as an endogenous synchronizer, melatonin provides the fetus/neonate with a remarkable arsenal of anti-inflammatory, antioxidant, and mitochondria protective capabilities while also carrying neurotrophic potential. The external disruption of circadian melatonin transfer to the newborn leads to an altered melatonin amplitude and acrophase timing with an inherent risk of developmental impairment.

**Figure 2 nutrients-16-01422-f002:**
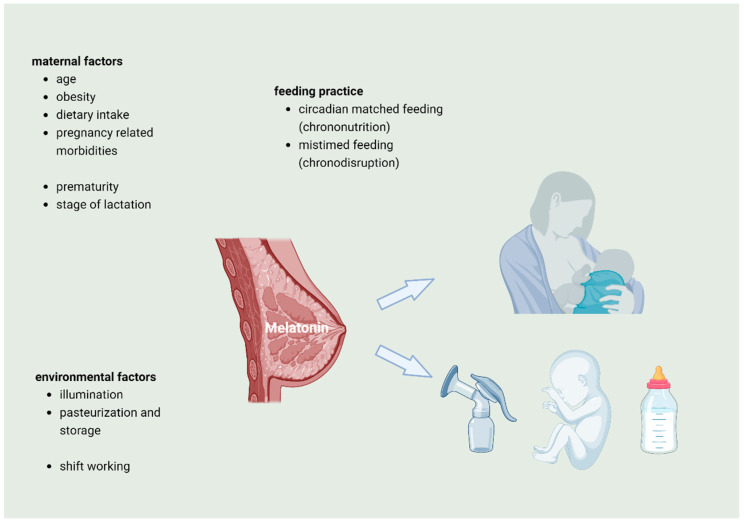
Variables that may influence melatonin concentration in human breast milk: maternal factors, environmental factors, and feeding practice as well as gestational age may influence the melatonin concentration during the different stages of lactation.

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
