# Peer review of "Melatonin in Human Breast Milk and Its Potential Role in Circadian Entrainment: A Nod towards Chrononutrition?"

_nutrients, 2024, doi:10.3390/nu16101422_

Round 1
Reviewer 1 Report
Comments and Suggestions for Authors
Recommendations:
1)In the Abstract the authors state that the human breast milk constitutes a unique natural source of melatonin, I suggest changing the word "unique" to "main": ...main natural source.. etc
Because, it is known in the literature that during an immunological response, macrophages can produce melatonin in an autocrine way, that is, although there is no production by the pineal gland, there may be production of melatonin.
2) Insert information about the immunomodulatory role of melatonin, at the beginning of page 2, where the authors describe:
Besides influencing circadian rhyrhmicity, melatonin acts as a free radical scavenger with anti-apoptotic, anti-inflammatory and mitochondrial stablizing functions (3-5).
Melatonin plays a role in immunological protection, including in colostrum, highlighting the importance of this hormone against childhood infections.
3)Insert citations from studies that validate the information cited in the following excerpt: "these factors include maternal age and condition which may critically impact neonatal melatonin supply via breast milk. Additionally, external risk factors such as shift work and bright light exposure during nurturing at night should be minimized to ensure the development and maintenance of a proper circadian entrainment of the newborn after birth.(page 2).
4) In topic three, it is necessary to add a brief paragraph illustrating the role of melatonin in colostrum compounds. In the literature, there are studies that address the immunological protection potential of melatonin, including that it has the potential to increase the protection of colostrum phagocytes, including during pregnancy. impacted by obesity, in which there is a reduction in melatonin in the colostrum of obese women.
The same for the topic 4, also illustrate the importance of melatonin from breastfeeding for the immunological protection of pre-term babies, f possible, include in the graphic diagram the information that melatonin also promotes immunological protection.
5) Page 7, check the typing of the text at: (summarized in (65)):
6) In the Topic "4.1. Maternal factors", the authors can also include the information that maternal obesity can also alter colostrum melatonin in obese women, although there are few studies in the literature on the subject, it is known that melatonin protects facing the development of obesity.
7) It would also be interesting to include a short text in the conclusion, or wherever the authors deem appropriate, highlighting that it is essential to encourage breastfeeding on demand, so that babies are breastfed at any time according to their needs, including at night.
Reviewer 2 Report
Comments and Suggestions for Authors
The authors provide an overview of an extremely relevant topic on which there is unfortunately very little well-informed and scientifically contrasted information. The importance of breastfeeding is paramount to human health, therefore this topic is of utmost importance. A quick search of Pubmed for literature revealed that there is very little information compared to many other topics, hence this overview is quite relevant.
Major comments:
1. The abstract does not reflect the relevance of the article, it should be completely rewritten. Consider the following three suggestions:
2. Most of the abstract (the first nine lines) refers only to the very first part of the introduction of the review. It should be shortened.
3. The second part of the abstract would be much better if it summarized the three main parts of the review in three sentences.
4. The last sentence of the abstract lacks power. A stronger sentence is needed here.
5. For section 3, a figure showing the diurnal/nocturnal variation of melatonin levels in: a) non-pregnant women, b) pregnant women (at different stages or weeks), c) postpartum and during lactation would be very useful. In addition, the melatonin concentration could be indicated at each stage.
6. A Figure summarizing the factors that influence the melatonin content in the composition of human milk would be great.
7. It would be useful to include throughout the text some melatonin concentrations in human breast milk that varies in different situations, comparing with normal levels.
8. Since the melatonin content in breast milk depends on the time of day, mothers who extract their own milk to feed their babies should pay attention to what time they extract the milk and give it to their babies at the same time on a different day. Right? So extraction during the day would be better for feeding during the day and extraction at night for feeding at night. It would be interesting to include a comment on this aspect in section 3. Although this aspect is addressed later, in section 4.2.1, it would be better if this aspect was also mentioned earlier, in section 3, and explained in more detail later, in section 4.2.1.
9. Is there clinical management for melatonin? For example, for pregnancy-related inflammatory conditions such as gestational diabetes, pre-eclampsia, miscarriage, fetal growth restriction or premature birth. It would be nice if a short section on these clinical aspects could be included in the review.
Minor comments:
There are several typo errors along the whole manuscript, such as: “hgher values”, “circadian regulation”, “coinicides with”, “sub-seuqently”, and so on… please double check completelly.
Comments on the Quality of English Language
There are several typo errors along the whole manuscript, such as: “hgher values”, “circadian regulation”, “coinicides with”, “sub-seuqently”, and so on… please double check completelly.
Reviewer 3 Report
Comments and Suggestions for Authors
I must unfortunately point out that this article, in addition to similar articles in recent years, does not present any new information or ideas. Moreover, in mirroring the errors of certain analogous publications, they have presented notions concerning newborn circadian rhythms as though they were supported by definitive scientific evidence, despite being rooted in a scanty pool of studies, thereby perpetuating the same fallacy. Authors should avoid emphasizing ideas lacking clear scientific evidence. Such statements may mislead many readers, especially young researchers unfamiliar with the subject matter.
Some major remarks:
P2, Paragraph 2: Above that, the initiation of pineal melatonin secretion by the newborn is absent for the first 2–3 months after birth (8,9).
P3, Parg2: Since the fetal pineal gland does not produce melatonin (8,33), the fetus is dependent on maternal melatonin, which easily crosses the placenta to enter the fetal circulation (34).
Comment: There is no evidence that the fetal pineal gland does not produce melatonin. In cited studies, only serum melatonin or urinary melatonin metabolites were measured. These studies demonstrated only no circadian melatonin secretion in newborn infants. However, this does not constitute evidence that the pineal gland cannot produce melatonin. Although many authors have suggested this in their reviews repeatedly, there has been no molecular analysis of the fetal pineal gland so far. Moreover, some studies demonstrated that serum melatonin concentration in newborn infants in the first days of life exist (for example,published in Clinical biochemistry 43.10-11 (2010): 868-872), reported an interquartile range of serum melatonin concentration of 8-26 pg/ml. This statement contains an ambitious expression; it could be argued that melatonin may not be produced sufficiently. However, it is also a sparking discussion. The optimal melatonin concentration for newborns remains unknown.
P3, Parg3:
Section “3. Circadian rhythmicity after birth”: The loss of placentally-transferred melatonin in the fetus is compensated mainly by its presence in breast milk (5,16).
This information represents just one hypothesis; the melatonin concentration in human milk may not sufficiently compensate for placental transfer of melatonin. Considering the low bioavailability of oral melatonin, a significant increase in serum melatonin concentration by breastfeeding is highly improbable. Additionally, references 5 and 16 share the same authors, hence only one should be cited.
“The higher concentration of melatonin and melatonin precursors may play an important role in fetal sleep promotion (24).”
Did authors really mean fetal sleep here? I think there is a mistake, because the subject of section and reference given here is related to breast milk.
P4, first 2 paragraphs: Here, the main purpose of the authors is not clear. Suddenly transitioning to the topic of microbiota, followed by the central nervous system (CNS), they discuss an experimental study unrelated to breast milk (Ref 55), and then attempt to link a study detailing the positive effects of breast milk on neurological development with melatonin.
Figure 1: Please change ”Circadian fluctuations in breast milk” as “Circadian fluctuations of melatonin concentration in breast milk”. Moreover, there is no study or scientific evidence indicating that melatonin in breast milk affects white and gray matter volume, cell protection or neurogenesis etc. Thus, the reader is misled here.
Page 6, parag 2 in section 4.2.1 light: Firstly, it is not correct to discuss the effect of maternal shift work on breast milk melatonin and its impact on neonatal circadian development. The neonatal period spans the first four weeks of life. Except for rare cases, no recently delivered mother works night shifts.
Secondly, it is once again confusing to discuss the melatonin content in breast milk and the development of the circadian system in the "fetus."
Page 8: It is not exactly correct to comment as “Synchronization to the external environment after birth with maintenance of circadian rhyhtm is mainly triggered by exposure to melatonin provided by breast milk.”. There is no scientific evidence in this issue.
Round 2
Reviewer 2 Report
Comments and Suggestions for Authors
The authors have conveniently addressed the questions and comments, consequently the manuscript has been improved in precision.
Reviewer 3 Report
Comments and Suggestions for Authors
Thank you greatly for revising the manuscript according to the reviewer's recommendations. The revised version is appropriate for publication in its current form.